# Methods for Measuring Frost Tolerance of Conifers: A Systematic Map

**Anastasia-Ainhoa Atucha Zamkova \*, Katherine A. Steele and Andrew R. Smith** 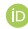

School of Natural Sciences, Bangor University, Bangor LL57 2UW, Gwynedd, UK;
k.a.steele@bangor.ac.uk (K.A.S.); a.r.smith@bangor.ac.uk (A.R.S.)
* Correspondence: afpa8a@bangor.ac.uk

**Abstract:** Frost tolerance is the ability of plants to withstand freezing temperatures without unrecoverable damage. Measuring frost tolerance involves various steps, each of which will vary depending on the objectives of the study. This systematic map takes an overall view of the literature that uses frost tolerance measuring techniques in gymnosperms, focusing mainly on conifers. Many different techniques have been used for testing, and there has been little change in methodology since 2000. The gold standard remains the field observation study, which, due to its cost, is frequently substituted by other techniques. Closed enclosure freezing tests (all non-field freezing tests) are done using various types of equipment for inducing artificial freezing. An examination of the literature indicates that several factors have to be controlled in order to measure frost tolerance in a manner similar to observation in a field study. Equipment that allows controlling the freezing rate, frost exposure time and thawing rate would obtain results closer to field studies. Other important factors in study design are the number of test temperatures used, the range of temperatures selected and the decrements between the temperatures, which should be selected based on expected frost tolerance of the tissue and species.

**Keywords:** conifers; frost tolerance; gymnosperms; freezing

## 1. Introduction

The ability of a plant to withstand freezing temperatures without suffering unrecoverable harm is known as frost tolerance. The frost tolerance of a plant can be modified by cellular processes that decrease susceptibility of cell damage to freezing temperatures, frost hardening. Damage can be caused by ice formation, leading to cell wall damage [1,2], protein denaturation [3], and cell and chloroplast membrane damage [4]. Frosts can also cause damage via phenomena such as photoinhibition, which occurs where plants are exposed to high intensity light energy in freezing conditions that result in photosystem II (PSII) being unable to discharge the excess of energy [4]. This results in the degradation of PSII, causing damage to chlorophyll. Although rarer in conifers than in other species [5], the combination of frosts with drought (or a frozen ground, which complicates the absorption of water by the roots) can lead to freeze–thaw embolism, thus increasing frost damage [6–8].

Gymnosperms, which tend to be evergreen, need to develop mechanisms to deal with frost damage differently from angiosperms, since they cannot use the strategy of shedding vulnerable tissue during cold times. Different plant tissues vary in their tolerance to cold temperature, and in conifers, the tissue of needles tends to have a lower frost tolerance than stem tissue [9,10].

An extensive number of academic reviews are available on the nature of frost hardiness of plants in general [11,12], cereals [13], woody plants [14–16], trees [3,17] and the molecular mechanisms of frost hardiness [3,11,16,18]. Bigras and Colombo (2001) published a book on the frost hardiness of conifers [19] that includes a chapter describing methods used for measuring frost hardiness [20].

We have only found three compilations of techniques used to measure frost tolerance in gymnosperms, a Canadian Forest Resource Development Agreement (FRDA) report by Keates (1990) [21], a review by Warrington and Rook (1980) [22] and a book chapter by Burr et al. (2001) [20]. Keates (1990) [21] focuses on sample selection, conditioning, freezing, testing and statistical analysis, while the review by Warrington and Rook (1980) [22] focuses on the techniques used for freezing and testing. The book chapter by Burr et al. (2001) [20] focuses on the description of the measurement techniques and their advantages and disadvantages. This 20-year-old chapter is the most up-to-date review and synthesis of frost tolerance techniques that compares each technique to one other and also analyses the technical details that make the techniques different from their idealized form.

There is wide variation in every aspect of study design, with different growing conditions, materials tested, freezing techniques and measurement techniques. The type of growing condition used would depend on the objective of the study. Frost tolerance is measured to evaluate many different things, such as the effect of environmental factors on frost tolerance [23–25]; the correlation with other physiological or phenological traits [26–28]; the differences in frost tolerance between different provenances, varieties, families and species, including ranking them by their frost tolerance [29–31]; the genetics of frost tolerance [32,33]; the rates of seasonal change in frost tolerance [34–37]; the mechanisms of frost tolerance [38,39]; and the effectiveness of the different frost tolerance measuring techniques [40,41].

The main techniques used for assessing frost tolerance were thoroughly described in the book chapter by Burr et al. (2001) [20], with their advantages and disadvantages. The main technique is simple visual assessment (VA), which consists of observing plants for signs of damage. Electrolyte leakage (EL) is a technique that is based on the measurement of changes in electrolyte levels due to the leakage of cellular content from damaged tissues into the surrounding environment. It consists of placing a treated sample in pure water and measuring the change in conductivity. The level of conductivity was compared to a control, and sometimes the sample was autoclaved in the water to make sure all the electrolytes had leaked [42,43]. Chlorophyll fluorometry consists of measuring the in vivo fluorescence of chlorophyll and the effects of freezing on chlorophyll [44]. Differential thermal analysis (DTA) consists of measuring exotherms during the freezing process and comparing them to a dead control [45]. Electrical impedance spectroscopy (EIS) is based on the reduced extracellular resistance caused by freezing [46] and the measurement of the electrical impedance of the tissue.

The primary goal of this review was to provide the information necessary to design a study that measures frost tolerance. The objectives were: (i) to document which techniques were used and how they were used; (ii) to document the technical constraints faced when measuring frost tolerance; and (iii) to note any reported correlations between different techniques in terms of results, by examining studies that use more than one method in further detail.

## 2. Materials and Methods

### 2.1. Search Strategies

The peer reviewed literature search was conducted using 'topic' for a basic search in Web of Science (Clarivate Analytics, Philadelphia, PA, USA), with the entire 'all years' (1970–2020) available time span and 'keywords' for a basic search in the Cab Direct database (CAB International) on 20 November 2020, which includes articles between 1968 and 2020. The search used the terms outlined in Table 1. No additional attempt at retrieving grey literature (evidence not published in commercial publications) was made.

**Table 1.** Definition of the main components of the search and the search terms used.

| | Definition | Search Terms [1] |
|---|---|---|
| Population | All gymnosperms | All gymnosperm Latin species names: (*Cycas OR Dioon OR Bowenia OR Macrozamia OR Lepidozamia OR Encephalartos OR Stangeria OR Ceratozamia OR Microcycas OR Zamia OR Ginkgo OR Welwitschia OR Gnetum OR Ephedra OR Cedrus OR Pinus OR Cathaya OR Picea OR Pseudotsuga OR Larix OR Pseudolarix OR Tsuga OR Nothotsuga OR Keteleeria OR Abies OR Araucaria OR Wollemia OR Agathis OR Phyllocladus OR Lepidothamnus OR Prumnopitys OR Sundacarpus OR Halocarpus OR Parasitaxus OR Lagarostrobos OR Manoao OR Saxegothaea OR Microcachrys OR Pherosphaera OR Acmopyle OR Dacrycarpus OR Dacrydium OR Falcatifolium OR Retrophyllum OR Nageia OR Afrocarpus OR Podocarpus OR Sciadopitys OR Cunninghamia OR Taiwania OR Athrotaxis OR Metasequoia OR Sequoia OR Sequoiadendron OR Cryptomeria OR Glyptostrobus OR Taxodium OR Papuacedrus OR Austrocedrus OR Libocedrus OR Pilgerodendron OR Widdringtonia OR Diselma OR Fitzroya OR Callitris OR Actinostrobus OR Neocallitropsis OR Thujopsis OR Thuja OR Fokienia OR Chamaecyparis OR Callitropsis OR Cupressus OR Juniperus OR Xanthocyparis OR Calocedrus OR Tetraclinis OR Platycladus OR Microbiota OR Austrotaxus OR Pseudotaxus OR Taxus OR Cephalotaxus OR Amentotaxus OR Torreya*). Additionally, ordinary names for the most common gymnosperms (*OR cedar OR celery-pine OR cypress OR fir OR juniper OR larch OR pine OR redwood OR spruce OR yew*). The common name for the largest division among gymnosperms (*OR conifers*), as well as the common name for conifer wood (*OR softwood*). |
| Trait | Frost resistance | Synonyms for frost (frost OR *freezing OR subzero OR cold *), joined with synonyms for resistance (*toleran * OR hard * OR resistan **), joined by the AND Boolean operator. |
| Technique/Method | Techniques used to measure frost resistance | Synonyms for techniques and technologies (*test * OR technique * OR measure * OR treat * OR trait OR analys **) |

[1] Separate strings in brackets were joined by the AND Boolean operator. * Asterisk wildcard was used to match words with different endings or beginnings (e.g., toleran* would match with tolerance, tolerant, etc.).

The search strategy was optimized during a scoping phase, which tried to find an appropriate balance between depth (number of papers found) and specificity (how well the papers matched the search criteria). This was achieved through an exploratory search (Table A1). The search terms were given a broader range by using the asterisk wildcard, which enabled matching a word with multiple beginnings or endings. Search terms were concatenated using the Boolean operators 'AND' and 'OR'.

Papers were accessed through Bangor University's library services and through green open access literature. No additional effort was made to find inaccessible articles published before the year 2010. Only English and Spanish language papers were included; other languages were discarded.

## 2.2. Article Screening and Inclusion Criteria

Literature search results were exported into Excel (Microsoft Corporation, Albuquerque, NM, USA), and duplicates deleted. Results were screened based on the inclusion/exclusion criteria listed in Table 2. Only original research papers that directly studied the measurement of frost tolerance of above ground tissues in gymnosperms were included (Table S1). Three rounds of selection were conducted. In the first selection round, entries were excluded based on the title, and the selection criteria in Table 2 were adhered to strictly, apart from ambiguity as to the species studied. In the second round, where the articles were included based on the abstract, the criteria in Table 2 were adhered to strictly. All reviews and modelling studies were excluded, and the abstract had to mention a gymnosperm species and frost tolerance measurements. Papers that studied species other than gymnosperms were included as long as they included at least one gymnosperm species

that had its frost tolerance studied. In the final selection round, selected papers that were available were excluded if they did not explain the technique used for measuring frost tolerance with sufficient clarity or detail.

**Table 2.** Inclusion and exclusion criteria for entries to be included in the systematic map (decided a priori).

| Inclusion Criteria | Exclusion Criteria |
|---|---|
| • Original research. | • Reviews, modelling studies, projections. |
| • Studies done on gymnosperms. | • No gymnosperm species studied. |
| • Directly measures frost tolerance. | • Uses indirect methods only to measure frost tolerance (e.g., DNA markers, amino acid levels, sugar levels, antioxidants) or does not measure frost tolerance. |
| • Performs measurements of damage on live, above ground tissue. | • Does not measure damage of above ground tissue (e.g., by measuring roots only) or uses dead tissue (wood, fossilised tissue). |
| • Uses holistic measurements, focusing on the organ/whole plant level. | • Focuses on only a specific part of frost damage (e.g., xylem embolism). |
| • Clearly explains what was measured and what species they used. | • Studies isolated cells instead of focusing on the organ level (e.g., cell cultures). Ambiguity or lack of clarity on the inclusion criteria. |

Screening criteria were decided after a discussion between AA, AS and KS. After several rounds of using the criteria for screening by AA and AS, they achieved 95% of coincidence in a sample of 50 titles with the criteria in Table 2.

### 2.3. Coding of the Articles

Metadata from all included research papers were recorded in an Excel workbook, with columns including basic publication data available (i.e., year, title, publication, DOI, language).

Information was extracted from the papers on the basis of three main categories: source and conditions of original biological study material (species, growing conditions, tissue studied); the treatment given (i.e., how freezing treatment was conducted, how thawing was handled, the temperature treatments used, the length of the treatment and its accuracy); and the measurement technique used (i.e., VA, EL, DTA, fluorometry and others). For studies where more than one technique was used, information on the correlation between the results of the techniques was also noted when mentioned. On the rare occasion when a field assessment was performed in natural conditions, this was noted. Equipment used for freezing tests was classified into categories according to its functionality and technology employed.

The categories for coding were decided a priori based on experience and practice with frost phenotyping methods. Examples of the extracted data files can be viewed in Supplementary Table S1.

The type of organ measured was the one noted, not the part of the plant on which freezing tests were performed. Plant growing conditions were classified according to the level of control exercised by the researcher, sometimes including categories with a wide range of variability. Thus, both pot-grown seedlings placed outside, irrigated and non-irrigated fields and old-growth forests were scored into the 'field' category.

Studies that were performed in the field were noted, and in the cases where the datasets were compared with results obtained in the laboratory, this information was used to verify that artificial freeze testing correlated with the desired characteristic.

### 3. Results

#### 3.1. Summary of the Evidence

In total, 3095 publications were found, of which 677 duplicates were deleted (Figure 1, made using the template by Haddaway et al. (2018) [47]). After screening by title, 495 articles passed the inclusion criteria. Of the 400 that were included after examining the

abstracts, 46 were unavailable (10 were not accessible through Bangor University's portal, and 36 could not be found online in full text version), and 42 were non-English language studies (nor Russian or Spanish, languages AA is familiar with). In the third round of selection, performed during scoring, which involved reading the full text of the paper, 283 studies that included all types of original published research were selected and scored.

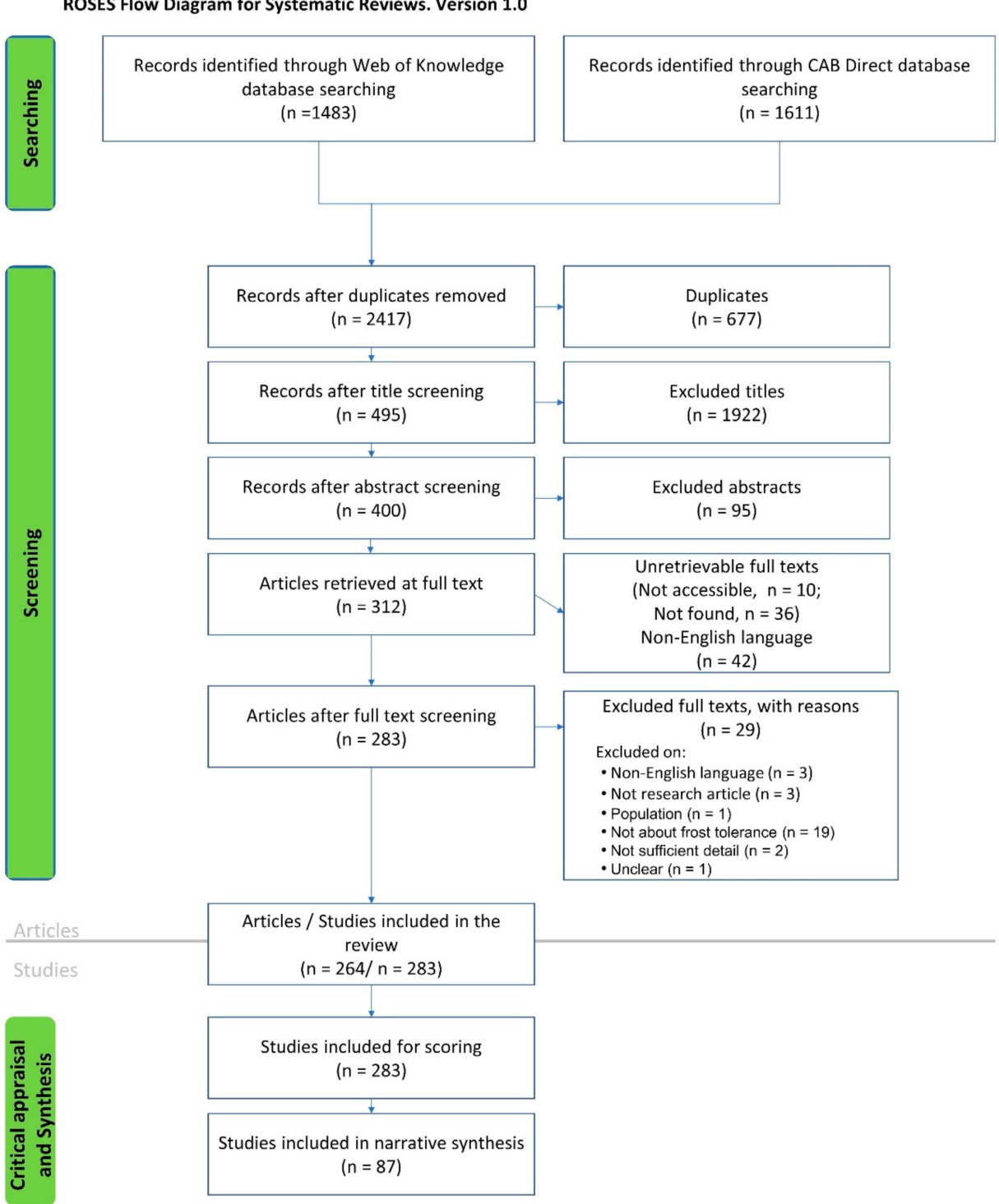

**Figure 1.** ROSES diagram outlining the search, screening and critical appraisal stages. Adapted from Haddaway et al., (2018) CC by 4.0 2018 [47].

### 3.2. Overview of the Included Articles and Studies

The 283 studies included in this systematic map were journal articles (n = 264, from 70 journals), conference proceedings (n = 10), notes in journals (n = 5), professional forester organization bulletins (n = 1) and research theses (n = 3) (Figure 1). The journals that published the largest number of articles were *Canadian Journal of Forest Research* (n = 49), *Scandinavian Journal of Forest Research* (n = 31), *Tree Physiology* (n = 24), *New Forests* (n = 12), *Forest Ecology and Management* (n = 11) and *Physiologia Plantarum* (n = 11). The rest of the journals were represented by <10 articles.

Most of the research on frost tolerance was published in the 1990s, with subsequent decline in the posterior decades (Figure 2). The techniques described in this review were old, with little change in the methodology used in the papers published after 2000.

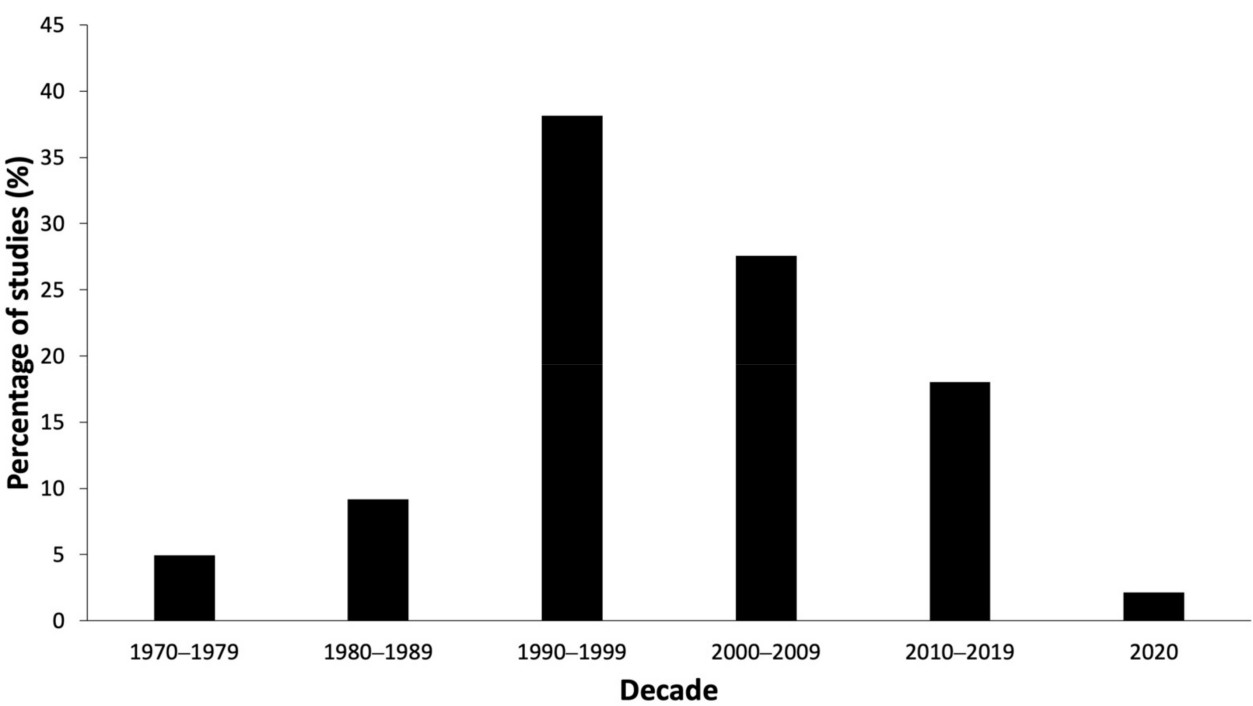

**Figure 2.** Distribution of selected papers by decade.

### 3.3. Key Findings

#### 3.3.1. Sample Selection

Among gymnosperms studied for frost tolerance, Norway Spruce (*Picea abies* (L.) H. Karst.) was the most studied species (*n* = 56), Scots Pine (*Pinus sylvestris* L.) the second (*n* = 50) and Douglas fir (*Pseudotsuga menziesii* (Mirb). Franco) the third (*n* = 46) (Table A2). Overall, spruces were the most studied genus among the studies included in this systematic map.

Most studies focused on 1–4 species, with only 10 studies researching the frost tolerance of more than five gymnosperm species. These studies were designed to measure the frost tolerance of either species that belonged to the same taxonomic group [48–51], or species that belonged to the same geographical area [12,52,53].

In some cases, freezing tests were performed on whole branches, and measurements were done on separate organs [54,55], or the tests were performed on whole plants, with measurements conducted on separate organs [56,57]. In field tests, assessment was also occasionally performed on separate organs [58]. However, freeze testing of individual needles [59], stems [60,61] and buds [62] also occurred. Studies performed measurements on freeze treated cut branches, whole plants and needles (Figure 3) on their own or in combination. Needles were the most studied individual organ.

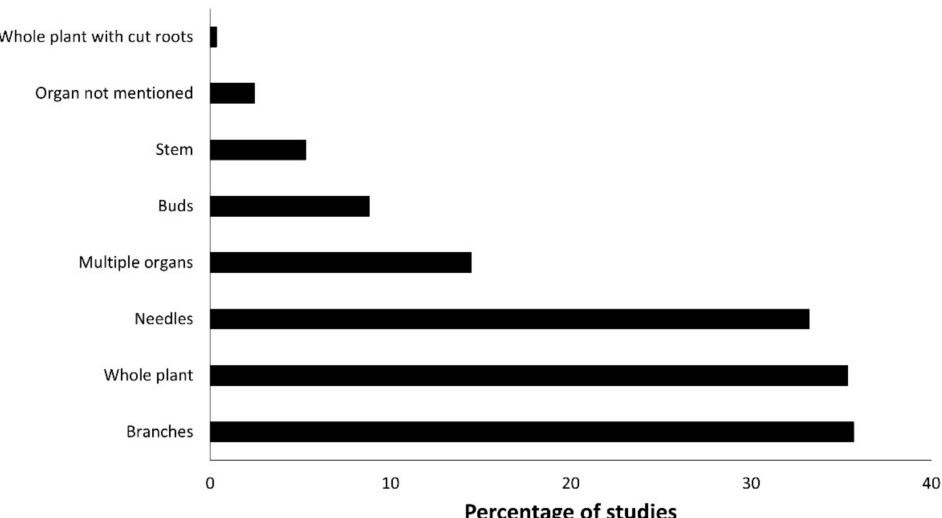

**Figure 3.** Plant organs studied for frost tolerance ordered by percentage of studies.

### 3.3.2. Pre-Conditioning

As can be seen in Figure 4, field-grown samples were most common, followed by greenhouse and growth chamber grown samples. Many studies tested the effect of different growing conditions, growing plants in different conditions for comparisons [63–65]. In some cases, equipment such as open top chambers used in the field, were used to control the air around the plant. Indoor growth rooms that allowed for the complete filtration of air, phytotrons, cold storage, tunnels and indoor rooms were much rarer (Figure 4).

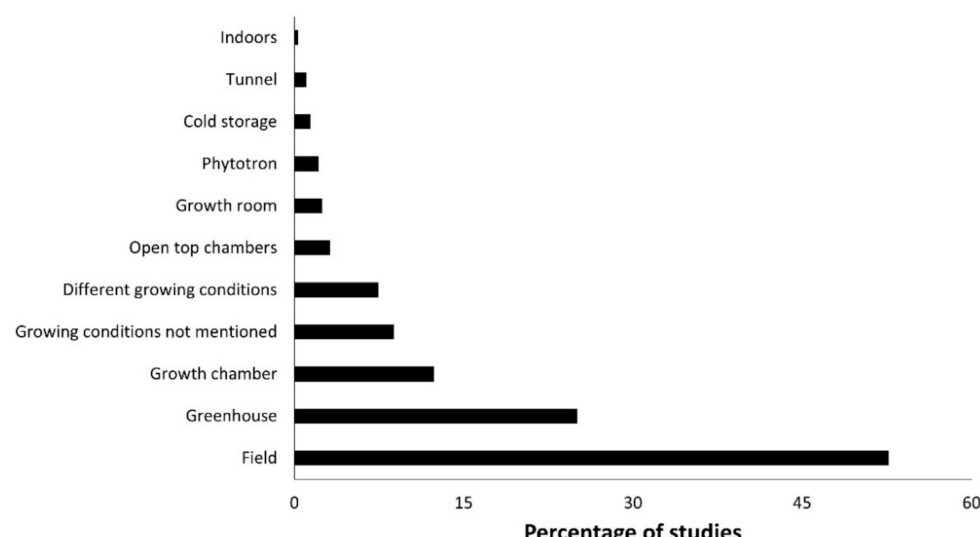

**Figure 4.** The conditions in which the experimental material was grown, ordered by percentage of studies.

### 3.3.3. Freeze Testing
Freeze Testing Equipment

Field testing

Of the 16 studies that did direct field observations, 11 (Table 3) measured material frozen in the field, without any freezing experiments. Most of them conducted a visual assessment, while two of them collected field material for assessment by EL. Seven studies performed both field and laboratory testing, only three of which correlated results between the two.

Some studies combined field observations with laboratory testing and measuring (Table 3), either checking for correlation between the two or not checking for correlation.

In the four studies that did both field observations and laboratory assessments but did not check for correlation, two performed the observations separately, with the same plants but not providing any information that allowed the comparison of the results for the same plants [49,66]. Another study was about freezing tolerance of three species of tree used as Christmas trees, white spruce (*Picea glauca* (Moench) Voss), balsam fir (*Abies balsamea* (L.) Mill.) and Douglas fir (var. *glauca* (Beissn.) Franco), which were subjected to indoor conditions for 10–20 days and later grown outside, meaning frost tolerance in the lab and field was measured at different times [67]. The study by Hodge et al. (2012) [51], while not explicitly measuring correlation between field and laboratory results, found that the ranking of species coincided in both methods.

In a study on different populations of *Pinus oocarpa* Schiede ex Schltdl., field observations obtained by visual scoring were correlated with laboratory-based EL measurements ($r^2 = 0.79$; if control excluded, $r^2 = 0.32$, no significance stated) [68]. The authors noted the importance of using a large sample when performing artificial freezing tests, as correlation between field observations and laboratory-based EL measurements was poor for the smaller groups, particularly families vs. provenances.

In a study on red spruce (*Picea rubens* Sarg.), field observations were done after establishing the level of frost hardiness of field-collected samples in the laboratory by EL, and observed damage in the field was strongly correlated ($r^2 = 0.61$, no significance stated) with the EL measurement [69].

Controlled enclosure testing

According to Warrington and Rook (1980) [22], there were two types of controlled enclosure tests that depended on the equipment they used: cold rooms and freezer cabinets (divided into laboratory units, field units and liquid nitrogen-based systems) and controlled environment rooms (divided into radiation frost chambers and advective frost enclosures).

Modifications to Non-Programmable Equipment That Allowed to Control Freezing Rate

The most frequently used type of equipment was the freezer in non-programmable and programmable versions (Table 4). Many modifications were used with freezers to control the rate of freezing, in some cases even outright modifying the freezers themselves. In other cases, while the freezer itself was not modified, additional equipment was used to control the rate of freezing, such as programmable controllers or cyclic timers. In some cases, plants were placed within insulating material to slow the rate of freezing, such as insulated boxes, a Styrofoam chest, plywood boxes, vacuum flasks or aluminium foil. Additional materials were used to provide a more spatially even temperature, such as aluminium shelves. When freezing to a lower temperature than the freezers could reach was needed, liquid nitrogen was used.

The studies that used freezing chambers (most non-programmable, half as much programmable) did not use advective frost enclosures or radiation frost chambers (Table 4) but simply the mechanism of freezing air. Some of the non-programmable units had modifications that allowed for the control of the rate of freezing, such as a Conviron, a programmable fan, a temperature controller or an external alcohol circulating system.

The third most frequently used technology (Table 4) was liquid baths. Due to water's freezing temperature of 0 °C, other liquids were used to provide sub-zero temperatures. The most frequently used liquid in order of number of studies was ethanol, methanol or an unspecified alcohol. Separate cases of use of polyethylene glycol, glycol, ethylene glycol, an ethanol:water solution and antifreeze solvent were noted.

**Table 3.** Studies that included field observations.

| | Frost Tolerance Measurement | | | | Organ Tested | | | | |
|---|---|---|---|---|---|---|---|---|---|
| | Total Studies | EL | Visual | Fluorometry | Whole Plant | Branches | Needles | Buds | Measure Correlation |
| Field only | 11 | 2 | 9 | 0 | 7 | 1 | 3 | 1 | * |
| Field and lab testing | 2 | 2 | 2 | 0 | 1 | 2 | 1 | 0 | Yes |
| Field and lab testing | 4 | 1 | 4 | 2 | 3 | 2 | 1 | 0 | No |

\* Not applicable; correlation could only be measured when more than one technique was used in the same study.

**Table 4.** Studies according to the freezing equipment used.

| Equipment Type | Non-Programmable | Modifications [1] | Programmable | Listed |
|---|---|---|---|---|
| Cold room | 4 | | 2 | |
| Field chamber | 2 | | | |
| Freezer | 45 | 6 programmable controllers; 5 liquid nitrogen; 2 vacuum flasks; 2 modified freezers; 1 insulated box; 1 aluminium shelf; 1 cyclic timer; 1 Styrofoam chest; 1 plywood box; 1 aluminium foil | 46 | |
| Freezing chamber | 52 | 1 Conviron controlled; 1 programmable fan; 1 temperature controller; 1 external alcohol circulating system | 25 | |
| Liquid bath | 27 | | | |
| Not mentioned | 63 | | | |
| Other | 7 | | | 2 growth chambers; 1 precision BOD incubator; 1 portable freezing system; 1 refrigeration unit; 1 refrigerator |

[1] Modifications to non-programmable equipment that allowed to control freezing rate.

Other technologies, such as field chambers, cold rooms (more non-programmable ones than programmable ones) or growth chambers, were much rarer, whereas some equipment was only used in one study (Table 4). This includes a refrigerator, a precision biochemical oxygen demand (BOD) incubator, a portable freezing system and a refrigeration unit.

Equipment used was also classified into programmable and non-programmable versions. Programmable versions allow for the control of the freezing and sometimes thawing rate. A substantial proportion of the equipment used (Table 4) was programmable, but more of it was non-programmable. Many studies do not mention the type of equipment they use for freezing. The remaining few were field tests, which do not require any equipment.

Freeze Testing Regimes

Studies differed in their testing regimes, which affect the measured frost tolerance by freezing rate, frost exposure time and thawing conditions. Other factors, such as the temperature range used, the numbers of temperatures and the decrements between temperatures, affect the accuracy of measurements but not the measured frost tolerance.

Freezing rate

Freezing rates were measured or given in 75.2% of studies, and the results presented below only apply to those. Defined here as the rate of temperature decrease per hour (in $K \cdot h^{-1}$), the scoring ignored some edge cases.

Some studies ($n = 17$) first equilibrated the sample at $-2$ °C, from room temperature to $-2$ °C, so the ice would form slowly, and a different freezing rate below $-2$ °C was used. The rate scored was the one below $-2$ °C.

In some studies ($n = 8$), in addition to freezing treatments using freezers and other equipment, samples were immersed in liquid nitrogen as a positive control for freezing damage. This meant that the rate of freezing for the liquid nitrogen exposed sample, depending on sample size, would be of ~196 $K \cdot s^{-1}$, as the temperature would jump from 0 °C to $-196$ °C in a matter of seconds.

Some of the studies ($n = 19$) that used a broad range of temperatures sometimes used different freezing rates for different temperatures, using higher freezing rates for lower temperatures. This was done in a stepwise manner, first decreasing the temperature to a certain threshold at a certain rate and then increasing the freezing rate. An average of the freezing rates used was scored.

Most studies used a freezing rate slower than or equal to $-5$ $K \cdot h^{-1}$ (Figure 5), with a small proportion of studies using freezing rates faster than $-5$ $K \cdot h^{-1}$.

Frost exposure time

Frost exposure time was scored as the time the sample spent exposed to the desired air temperature. In some cases, it should be noted that larger samples, such as seedlings or large trees, will take a longer time to equilibrate with the air than smaller samples, but only the length of the air temperature exposure was noted, as the true value of the plant experiencing the temperature was not available.

The most frequently used exposure time was of 1 h (Figure 6), followed by the exposure time above 3 h and up to 4 h. Flash exposure, where samples were taken out when temperature in the freezer was reached, was the third most frequently used method. It should be noted that a few studies use different exposure times for different organs. Overall, the majority of studies use a frost exposure time up to 4 h.

Thawing rate

Thawing rate was only measured in 39.9% of the studies, with the most common thawing rate being 5 $K h^{-1}$, followed by 7–10 $K h^{-1}$, with 2 $K h^{-1}$ being the third most frequent (Figure 7).

It should be noted that the remaining 60.1% of studies did not mention the thawing rate used, as it was difficult to control. Different techniques were used to slow the thawing rate even when precise control was unavailable. Some studies (n = 9) used a stepwise procedure, where frozen samples were placed at temperatures until they equilibrated, at

several temperature decrements. This helped reduce the rate of thawing by reducing the temperature differential between the frozen sample and the surrounding temperature.

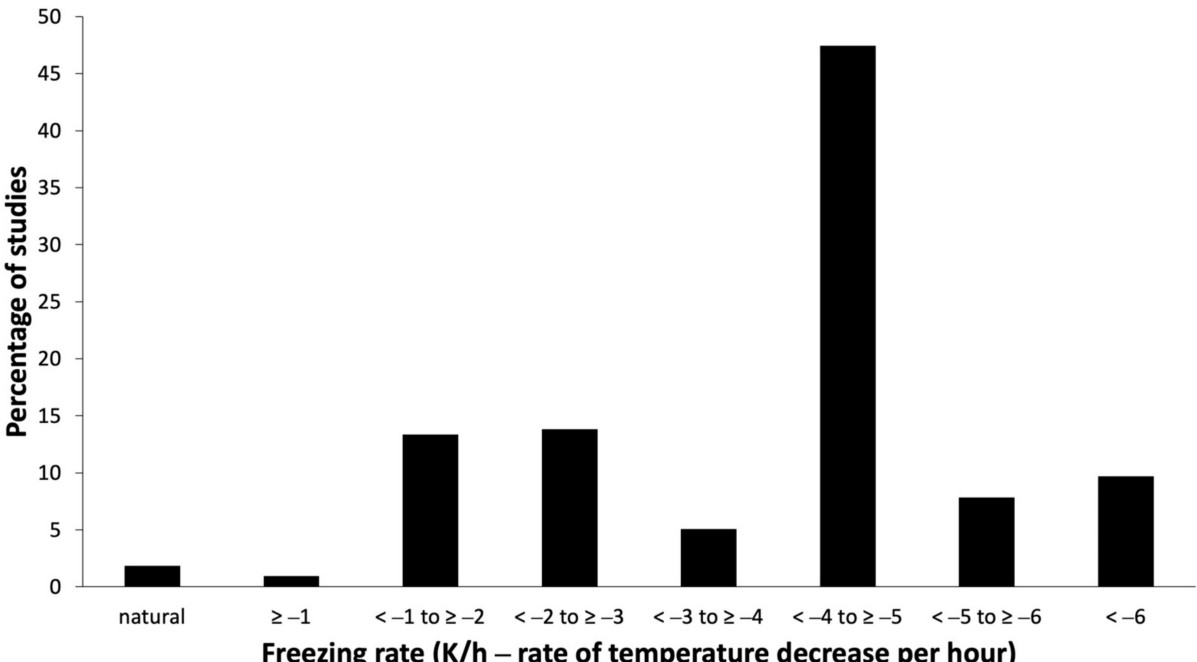

**Figure 5.** Number of studies binned according to the freezing rate used.

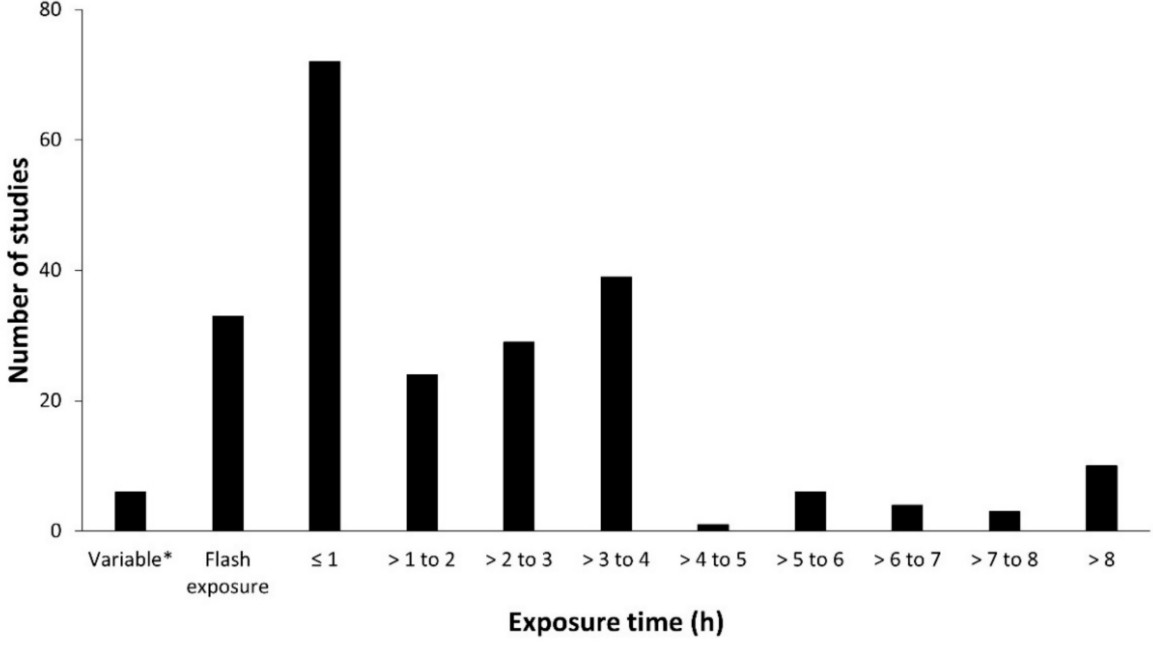

**Figure 6.** Number of studies binned according to the exposure time they used. * Uses different exposure times in the study.

In order to avoid the extremely high temperature differential between the frozen sample and ambient temperature, in most cases (n = 107), the sample was placed in refrigerators or other such freezing devices at temperatures between 0 and 5 °C before it was exposed to the much warmer ambient temperature.

In a minority of cases (n = 6), samples were left at warm ambient temperatures to warm.

Freezing temperatures

Temperature range (the difference between the highest and lowest test temperature used in the study) was scored for 33.9% of the studies. The most frequently used temperature range was of 10–19 K or below (Figure 8). Higher temperature ranges were much less frequent, but the ranges extended quite widely, with the highest temperature range being 196 K (the difference between 0 °C and −196 °C, the temperature of liquid nitrogen).

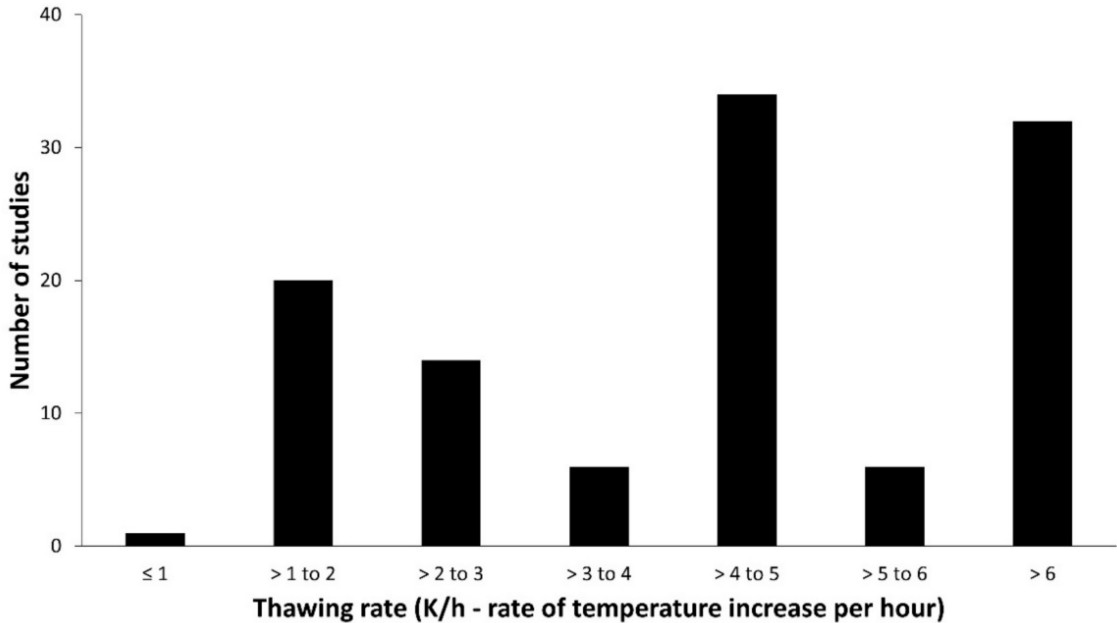

**Figure 7.** Number of studies binned according to thawing rate, among the studies for which the thawing rate is known.

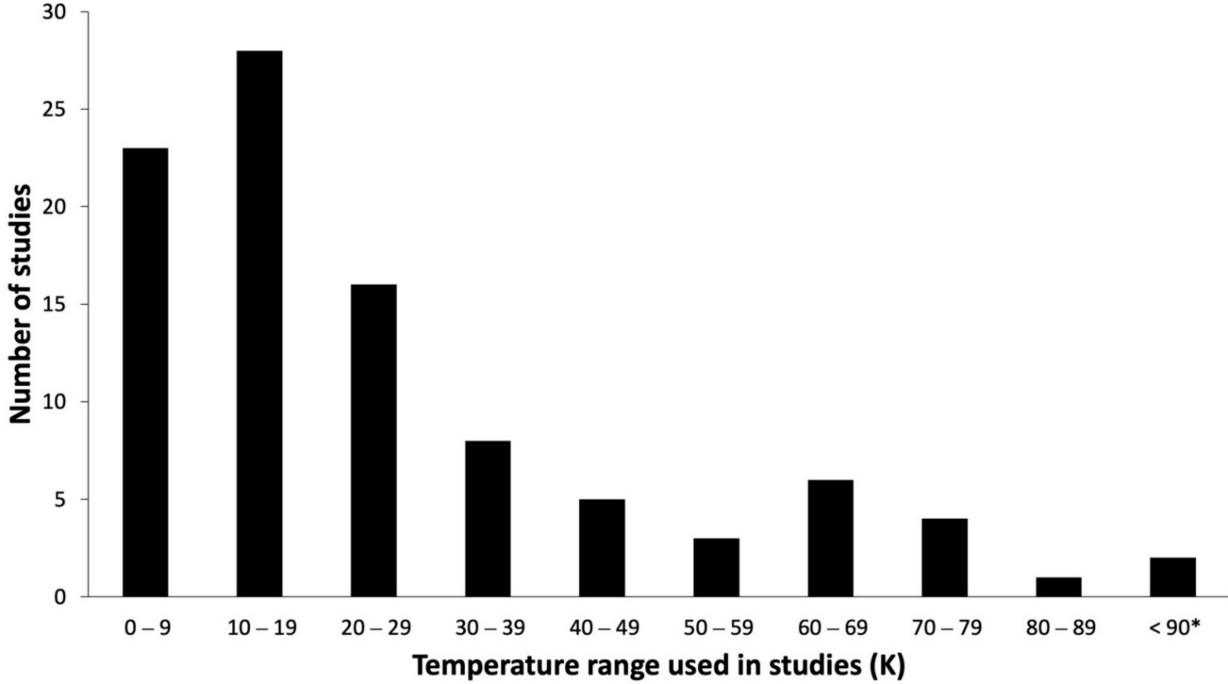

**Figure 8.** Number of studies binned by temperature range used (difference between the highest treatment temperature, excluding the negative control, and the lowest temperature). * Includes temperature ranges between 90–196.

The number of test temperatures was scored in 56.5% of studies. The most frequent type of study only used one test temperature (Figure 9). The second and third most common set ups involve the use of three and four test temperatures, respectively.

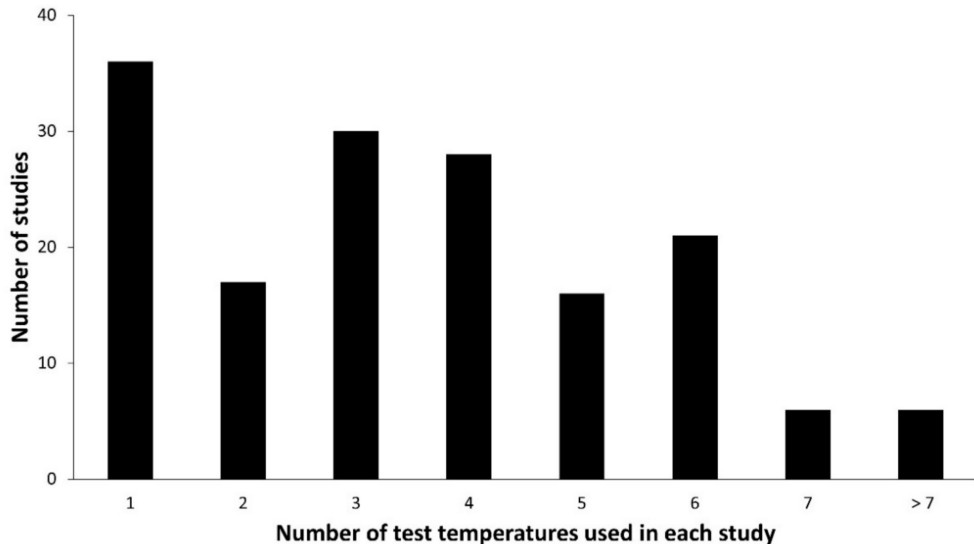

**Figure 9.** Number of studies according to the number of test temperatures used in each study.

Temperature decrements were defined as the smallest distance between two adjacent test temperatures used in a study. Temperature decrements of 1–2 K were quite frequently used in 20.6% of studies for which temperature steps could be calculated (Table 5). This value fell within the range of accuracy of reached temperatures (the difference between temperatures programmed and actual temperatures achieved), which was between 0.1 and 2.0 K for the studies where it was measured (Table 6).

**Table 5.** Number of studies according to temperature decrements (difference between two adjacent test temperatures) used in the studies. Table only includes those studies where the temperature decrements were given or could be calculated.

| Temperature Decrements | Number of Studies |
|---|---|
| 1 to 2 K | 13 |
| 3 K | 6 |
| 4 K | 10 |
| 5 K | 13 |
| More than 5 K | 10 |
| Different steps depending on temperature * | 11 |

* Use different temperature steps depending on the temperature, e.g., use a temperature decrement of 2.5 K between 0 °C and −20 °C, and temperature step of 10 K between −20 °C and lower.

**Table 6.** Number of studies by the accuracy of the achieved test temperatures (the difference between temperatures programmed and actual temperatures achieved), for the studies that give this value.

| Accuracy (K) | Number of Studies |
|---|---|
| 0.1 | 3 |
| 0.2 | 2 |
| 0.3 | 1 |
| 0.5 | 5 |
| 0.7 | 1 |
| 1 | 5 |
| 1.5 | 3 |
| 2 | 2 |

The majority of studies used temperature steps of 3 K or above (Table 5), which fell outside the accuracy range (Table 6).

### 3.3.4. Measuring Freezing Damage

Visual assessment (VA) was the most common method used for measuring frost damage (Figure 10). It is noteworthy that a large proportion of studies used more than one measuring technique, and as every instance of a measuring technique being used was counted, this resulted in counting the same study more than once (hence the total sum of percentages adding to more than 100%).

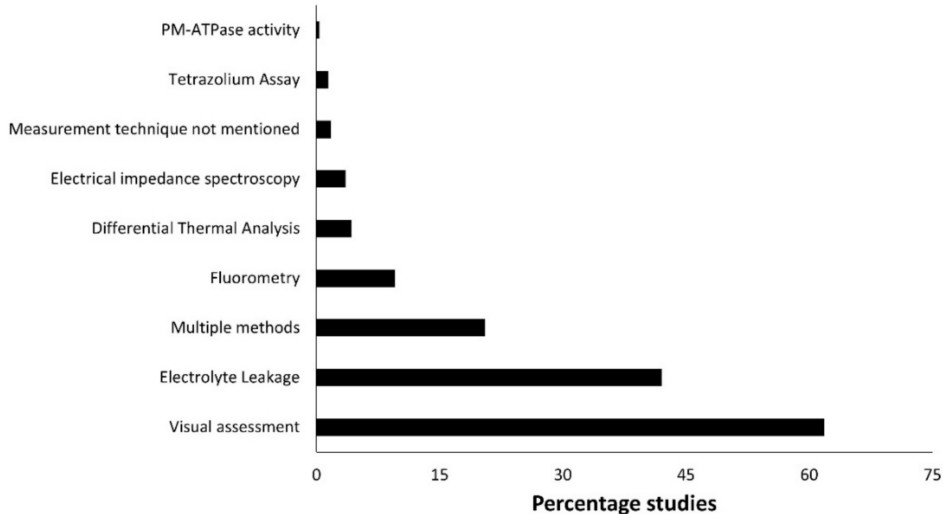

**Figure 10.** The frost tolerance measuring techniques used, by percentage of studies.

The second most used technique was EL (Figure 10). The third most used technique was fluorometry; differential thermal analysis, DTA, was fourth; and EIS use was marginal. Techniques such as the tetrazolium assay and the PM-ATPase activity measurement were rare and were not used after 2004.

A total of 58 studies (Figure 10) combined different measuring techniques, two of which were the previously discussed studies that combined field and laboratory measurements that checked for the correlation between the two [68,69].

The most common comparison was between VA and EL, the second most used technique (Figure 10), and they seemed to be well correlated, with statistically significant correlations, such as in studies in maritime pine (*Pinus pinaster* Aiton), $r^2 = 0.31$ in spring, $r^2 = 0.79$ in autumn, $p < 0.05$ [70]; only below $-30\,^{\circ}$C in Scots pine, $r^2 = 0.94$, $p < 0.0001$ [71]; and in a study of multiple pine species, $r^2 = 0.64$, $p < 0.0001$, [48].

The second most common comparison was between VA and fluorometry. Fluorometry was also highly correlated with VA, with statistically significant correlations, as shown in a study in multiple species ($r^2 = 0.85$, $p < 0.001$) [12]; in Aleppo pine (*Pinus halepensis* Mill.), $r^2 = 0.97$, significance stated but $p$-value not provided [72]; in exotic firs, $r^2 = 0.36$ for needles, $r^2 = 0.48$ for stems, and $r^2 = 0.21$ for buds, $p < 0.0001$ [49]; in maritime pine, $r^2 = 0.19$ in spring, $r^2 = 0.61$ in autumn, $p < 0.05$ [70]; and in Aleppo pine, $r^2 = 0.67$ at 200 h ($p$-value not mentioned, but significance stated) [73].

The third most common comparison was between EL and fluorometry. High degrees of statistically significant correlation were found in a study in maritime pine, $r^2 = 0.50$ in spring, $r^2 = 0.55$ in autumn, $p < 0.05$ [70].

VA and EIS measurements of frost hardiness in Scots pine were correlated, $r^2 = 0.95$, no significance stated [74], but no correlation was found in another study of Scots pine [75]. A study in Douglas fir found agreement between the ranking achieved by VA and EIS [76]. EL and EIS were found to be correlated ($r^2 = 0.91$, no significance stated) in a study of *Pinus bungeana* Zucc. ex Endl. [77].

In a three-way comparison between EL, DTA and VA done in ponderosa pine (*Pinus ponderosa* Douglas ex C. Lawson), Douglas fir and Engelmann spruce (*Picea engelmannii* Parry ex Engelm.), it was seen that, while the measurements agreed, EL was more precise than DTA, while the VA, which was performed following a whole-plant freezing test, was the least precise [41]. In another three-way comparison between EL, VA and fluorometry, an overall correlation of $r^2 = 0.85$ (no significance stated) was found for Douglas fir, white spruce, Engelmann spruce, contorta pine (*Pinus contorta* Douglas ex Loudon) and western larch (*Larix occidentalis* Nutt.) [78].

## 4. Discussion

### 4.1. Sample Selection

The most common species studied in the selected studies were some of the most economically important commercially grown species in Europe: Norway spruce and Scots pine being species native to Europe, and Douglas fir an introduced species [79].

Studies performed tests on either the entire plant (only with small plants) or part of the above-ground tissue. When considering partial components of a tree, the most common procedure was to take a branch cutting. Branch cuttings contain all the relevant organs: stem, needles and, depending on the timing, buds. Branch cuttings can be evaluated in their entirety, or each separate part could be evaluated on its own.

Branches have many advantages for sampling: they are small, they contain all relevant organs and cutting branches allows for measuring the tolerance of the same tree for different test temperatures. Understanding the whole-plant freezing resistance would be the objective in most cases, as the resistance of branches on its own does not inform the survivability of plants in the tested conditions. However, whole-plant freeze testing is inherently destructive, as freezing the plant is likely to kill it or damage it, complicating further tests on the plant. This means that a plant can only be used once when conducting using a whole-plant freezing test, while branch samples allow for a plant to be tested multiple times. Moreover, whole-plant freezing requires larger freezers and a longer freezing time, as the larger mass of the sample will take longer to equilibrate with the surrounding air. Roots also need to be protected. Freezing containerized plants [80,81] serves to protect the delicate roots, which are otherwise exposed to conditions that would not occur in nature, as they would be protected by the soil. The increased mass and volume of the container, however, would impose limitations as to the type of freezing equipment used. Strategies to slow down the freezing rate, such as using vacuum flasks, would be harder for whole plants due to size constraints. Despite the complications of whole-plant freeze testing, it was the second most common type, presumably because it allows for conditions that are closer to real-world field results.

Needles, unlike buds, which are formed in autumn and flushed in spring, are present throughout the year. They also lack the protection the stem enjoys, in the form of protective barriers such as the bark. Needles are also the most sensitive organ that was present year-round. Visual damage to needles is usually immediately visible, whereas stems and buds are harder to examine and frequently need to be cut for examination [9,82,83], although sometimes a superficial assessment is sufficient [84,85].

In general, the sample selected for freeze testing would depend on the availability of the biological material, the frequency of testing, the number of replicates for each biological sample, the objective of the study, the available freezing capabilities and the type of measurement used.

### 4.2. Pre-Conditioning

While Keates (1990) [21] assumes plant material will generally come from either field-planted stock or seedlings from nursery or greenhouse culture, this review found more variability in the sources of plant material collected.

The conditions under which plants were grown before or even during freeze testing depended on the goal of the measurement and were highly variable. The prevalence

of studies on field grown material suggests that the most important reason to test frost tolerance was the measurement of frost tolerance in real-world conditions, without the artificial constraints of the laboratory.

This review focused on the conditions under which the experimental material was grown immediately before or during the freezing tests. This was done because it was common to either grow or obtain seedlings from nurseries and then move them across different growing conditions as they grew [51,86–88] or in order to test the effect of growing conditions [89–92].

The pre-conditioning of the experimental design differed according to the aims of each study. For example, a study performed on indoor grown trees aimed to explore the frost resistance of indoors Christmas trees [67], whereas another study, performed with plants left in cold storage, aimed to observe the effect of cold storage (which is commonly used by commercial nurseries) on frost tolerance [93]. Another study used cold storage, aiming to measure the decrease in stored carbohydrates and their effect on frost hardiness [94].

Field conditions offer less control over growing conditions than every other type of pre-conditioning. It was thus the form most similar to natural conditions. Field conditions differ between each other on the level of control (for example, the level of watering and fertilization). Open top chambers allow growing plants in the field exposed to the same light, hydrological and temperature regimes as other field-grown plants while controlling the gaseous environment in which the plants were grown. This was done to measure aspects such as the effects of acid mist [69,95,96], ozone [97] or increased $CO_2$ [71].

Glasshouses allow for more control of growing conditions, providing heating, watering, and additional lighting when necessary. Some glasshouses also filter the air for particles, thus permitting control of air quality. Glasshouses rarely offer the possibility to cool beyond opening windows when outside temperatures and sunlight create heat stress conditions. While additional lighting can be provided, blackout darkness is rarely available in glasshouses. Humidity control beyond watering is also rarely available in glasshouses.

When control over every aspect of growing conditions is desired (temperature, photoperiod, light intensity, air composition and humidity), growth rooms and phytotrons would be used, which provide the ability to control every aspect of plant growing conditions.

Thus, growing conditions will depend on the objective of the study and the level of control over growing conditions necessary to achieve these objectives. As each additional level of control will require an additional cost, researchers should focus on the growing conditions that achieve their objectives in the most cost-effective manner.

### 4.3. Freeze Testing

#### 4.3.1. Freeze Testing Techniques

Keates (1990) [21] classified freeze testing into two types: field and laboratory testing. Warrington and Rook (1980) [22] classified freeze testing into three main types: field studies, controlled enclosure studies (equivalent to laboratory freezing according to Keates (1990) [21]) and temperature gradient bars. No studies that used temperature gradient bars were found with the search criteria used in this review.

#### Field Testing

The main difference, as both Keates (1990) [21] and Warrington and Rook (1980) [22] note, was that for the field tests, the results of naturally occurring frosts were observed, whereas in laboratory/controlled exposure studies, frosts can be controlled.

The simplest method of freeze tolerance measuring was to observe the results of naturally occurring frost events in field-planted stock. These observations were perceived by scientists and foresters to be the only real measure of frost hardiness [22]. However, as noted in both reviews by Warrington and Rook (1980) [22] and Keates (1990) [21], field testing has many limitations, with both reviews highlighting the unpredictability of field conditions. Warrington and Rook (1980) [22] reported that in some years, plants with different frost tolerances can be killed by a particularly harsh frost, and in others, none of

them would be harmed due to a particularly mild year. This problem can be accounted for by running the observations for a number of years in different sites, which increases cost.

The lack of precision of field testing was another problem, as measuring frost conditions across a site can be a very costly endeavour, due to microsite variation [22]. Effects of frosts would also be hard to distinguish from other effects of the site, such as drying winds or weed competition [21,22]. These problems could be overcome by increased replication, which is costly.

The high cost of field observation [21] could explain why studies that include field testing represented only 5.6% of the total number of studies reviewed. Field testing was rare, and the majority of studies were done in controlled enclosures, where frosts can be simulated on demand. Furthermore, as shown in the two studies that measured correlation between field observations and controlled enclosure results, field results were strongly correlated with controlled enclosure results [68,69].

Controlled Enclosure Testing

Keates (1990) [21] found that three main equipment types were used to administer freezing tests in the laboratory: freeze chambers, liquid baths or temperature gradient bars (classified by Warrington and Rook (1980) [22] into a different main category).

Although studies that do freezing tests using temperature gradient bars seemed to be important enough to put in a different category by Warrington and Rook (1980) [22], none of the studies included in this review use temperature gradient bars. This could be because temperature gradient bars were only suitable for extremely small samples [22]. The latest reference used by Keates (1990) [21] when talking about this technology was from 1983. This technology seems to be old and could have been abandoned as newer technologies became available.

Warrington and Rook (1980) [22] classify laboratory testing into two types: cold rooms and freezer cabinets and controlled environment rooms. Cold rooms can be lab or field based and use liquid nitrogen to cool the unit, and controlled environment rooms can be either radiation frost chambers or advective frost enclosures. Neither type of controlled environment room was found in this search, and they were not described in the later review by Keates (1990) [21]. Presumably, these were also old technologies that were abandoned as newer technologies became available.

Evidence from the Warrington and Rook (1980) [22] review suggests that as technologies improved, the techniques used before the 1980s were abandoned in favour of machines that could perform controlled freezing tests. The reason why radiation frost chambers or advective frost enclosures were abandoned is unclear, but they did not appear in any studies beyond 1978.

While programmable versions allow for more control over the freeze testing process, non-programmable freezers and freezing chambers were more widely available in most laboratories, as they were not specialist equipment. Thus, the wide use of non-programmable freezers cannot be used as an argument in their favour, since their widespread use was presumably due to their availability and cost rather than inherent technical advantages.

Liquid baths, while they do allow for a uniform freezing, have the problem of a limit to the coldest temperature achieved, as the liquid becomes solid. It was thus logical that equipment that relies on air freezing, which can achieve extremely low temperatures, was more common.

In the studies published since 2010, the majority that name equipment used a programmable freezer (34%) or a programmable freezing chamber (25%). Multiple studies by different groups used the Forma Scientific Model 8270/859M programmable freezer [82,83,98], while other labs used their own equipment.

In general, programmable specialized equipment can be presumed to better serve the purpose of frost tolerance measurement, despite the higher cost and the widespread availability of non-specialized freezers.

4.3.2. Freeze Testing Conditions

Freezing Rates

Freezing rates were an important factor for the assessment of frost tolerance. High freezing rates that can be artificially achieved are not expected to occur in nature, as large masses of air take time to cool. Thus, in order to measure frost tolerance that is closer to field values, freezing rates that are closer to natural ones should be used.

Freezing rates that can be achieved will heavily depend on the equipment used and the modifications made to said equipment (Table 4). For some types of programmable equipment, the freezing rate can be programmed, allowing for this factor to be controlled. However, the most used types of equipment were non-programmable (Table 4), and the rate of freezing could only be decreased to a degree by the use of insulation. Many studies, 22.2%, did not mention the type of equipment they used for freezing, thus inspiring doubts about the freezing rates they mention.

Higher freezing rates seemed to lead to a decreased frost tolerance temperature in Norway spruce buds, although the difference was only of 2.6 K [99]. It should be noted that this study only used freezing rates between $-1$ and $-5$ K·h$^{-1}$, not using rates faster than $-5$ K·h$^{-1}$.

In a study of Leyland cypress ($\times$ *Cupressocyparis leylandii* (A.B.Jacks. & Dallim.) Dallim.), a freezing rate of $-6$ K·h$^{-1}$ led to tip browning, while the slower freezing rates of $-4$ K·h$^{-1}$ and $-2$ K·h$^{-1}$ did not cause such damage, using the same freezing temperature [100]. In a study of radiata pine (*Pinus radiata*, D. Don) seedlings, higher freezing rates caused higher levels of damage across different treatments, maintaining temperature, thawing rate and frost duration constant [101].

The majority of studies scored in this review used a freezing rate of $-5$ K·h$^{-1}$ or slower (Figure 5), with most studies using a freezing rate of $-5$ K·h$^{-1}$. This could be because of the increased cost and time of slowing freezing rates from $-5$ K·h$^{-1}$ and the small effect on the measured results at freezing rates slower than $-5$ K·h$^{-1}$ [99]. However, it seems that significant efforts were made in multiple studies to ensure freezing rates slower than $-5$ K·h$^{-1}$ (Figure 5), as achieving such a rate would be more costly. Additionally, as shown in a study in Scotland, freezing rates faster than 5 K·h$^{-1}$ occurred very rarely in nature [102].

Frost Exposure Time

Frost exposure was an important factor in determining frost tolerance. Increasing the length of frost exposure significantly increased the rate of damage in radiata pine [101].

As longer exposure times are more time intensive, it is not surprising that the majority of studies scored used a frost exposure time up to 4 h (Figure 6), with a peak at 1 h.

In a study on frost duration in Iran, frosts with durations of 0–3 h represent 11.2–36.6% of all frost events in four sites, while frosts of 6–9 h make up 15.3–22.8% [103]. Frost duration, ranging between 0 and 24 h, had a skewed distribution, with a majority of frosts having a duration below 12 h (58.3–90.1%) in the four Iranian sites. Damage increases linearly with exposure time [101] between 2 and 8 h of exposure time, while the difference between lower exposure times was much higher, possibly non-linear [104].

Due to these non-linear effects of increased time in short duration frost events of less than one hour [104], higher levels of control and accuracy have to be used to ensure uniform conditions across the different tested samples, as slight differences in exposure duration can cause large differences in effect. Additionally, in addition to the more complicated setup, the duration of the frosts seems to span a wide range. Thus, researchers that wish to estimate the effects of frosts in the field should use frost duration times between 1 and 12 h, 0–12 h being the most frequent [103] type of frost in nature, and should avoid durations below 1 h due to the non-linear effects [104]. Frost durations of 1–4 h cause similar levels of damage, with less damage below 1 h [104]. If duration in the site where the tested plant would be planted is known, that duration should be used. In the absence

of such knowledge, a reasonable duration between 1 and 12 h should be used, and the same duration should be used in all measurements to allow for comparisons.

Flash exposure was still quite prevalent in these studies (Figure 6). Flash exposure was usually done by removing samples when the desired test temperature was reached. Its frequent use could be due to technical constraints; while programmable freezers can be programmed to reach and maintain a certain temperature, most non-programmable freezers can only be set to the lowest temperature setting they have. Thus, keeping samples in such a freezer would lead to a lower test temperature than the desired one, unless the test temperature was the lowest temperature the freezer could achieve.

Thawing Rate

Thawing rates are another factor that could affect the measured frost tolerance. Many studies did not measure thawing rates, although they used different thawing times. Higher thawing times would lead to slower thawing rates if the rate of thawing was uniform.

In a study of primordial shoots of Norway spruce, a slower thawing rate leads to less frost damage at identical frost temperatures [99]. The study notices large differences in measured frost tolerance between 2 and 18 h of thawing for the same temperature differential, with faster thawing leading to more damage. This relationship was exponential, with a threshold point of 6 $K \cdot h^{-1}$; above that threshold, frost damage increased exponentially in relation to thawing rate, whereas below that threshold, thawing rate has a linear relationship with frost damage [99].

Thawing rate increases from 2 $K \cdot h^{-1}$ to 10 $K \cdot h^{-1}$ also seem to lead to higher degrees of damage in a study in radiata pine, where freezing rates and frost exposure were maintained constant [101]. A study in Norway spruce by Floistad and Kohmann (2010) [105] found that increased thawing time (and slowing thawing rates) led to less freezing damage. However, it should be noted that the study compares a 16 h thawing time to a month-long thawing time.

Large masses of air take longer to warm than what can be achieved artificially by taking a sample from the freezer and putting it at room temperature. It is thus likely that studies that do not try to control the thawing rate will measure higher levels of frost damage than what they would be in field conditions.

Freezing Temperatures

While the freezing rate, frost exposure time and thawing rate used by researchers partly depend on the availability of equipment that allows for the control of these factors, researchers have more control over the choice of test temperatures, the number and range of temperatures compared and the decrements between the chosen test temperatures. Freezing rate, frost exposure time and thawing rate also change the estimated frost tolerance because it will be contingent upon them. Chosen test temperatures do not change the frost tolerance, whereas frost duration, freezing rate and thawing rate do. The range of temperatures used, the decrements and their number will allow for a more precise and accurate calculation of the frost tolerance.

Using a wide array of freezing temperatures allows for the calculation of the frost tolerance or the temperature at which 50% of the sample was damaged ($LT_{50}$, median lethal time), the middle value on the frost response curve. This simple method of calculating frost tolerance from experimental data is used in many studies [39,45,82,106]. Three elements will determine the accuracy of the estimated $LT_{50}$: the range of temperature used and whether it includes the real value of $LT_{50}$; the decrements between the temperatures, with smaller decrements allowing for a more precise value; and the number of temperatures used, as fewer values mean a higher level of freedom on the shape of the frost response curve.

In order to calculate the real frost tolerance, a range wide enough to include the frost tolerance value should be used. If the real value of frost tolerance falls outside the tested range, calculations of the frost tolerance value will be much less precise. Thus, in cases

where the real value of frost tolerance is unknown, a range as wide as possible should be used. It should be noted, however, that by using pre-tests, the approximate value of frost tolerance can be estimated, and a narrower range will still lead to informative results [107,108]. If an approximate value could be estimated from the literature or prior knowledge, a wide range of temperatures was also unnecessary. Most studies use a temperature range of 10–19 K or below (Figure 8). This means that most studies use a narrow window of test temperatures, and if the real value falls below or above the tested range, it would not be possible to estimate it.

It was common for only one test temperature to be used (Figure 9). This type of study, however, can only determine whether the real frost tolerance falls above or below this test temperature: if frost damage was above 50%, the test temperature was below frost tolerance, and if it was below 50%, it was above. This level of accuracy will be unsatisfactory, which was why the majority of studies use 3–4 test temperatures (Figure 9). However, as the use of each additional test temperature increases costs, there is a trade-off between accuracy and cost.

Temperature decrements determine how accurate the test will be, and smaller decrements will allow uncertainty to reduce, giving a better fit of the frost response curve. However, it should be noted that temperature decrements should be above the accuracy level of the freezing equipment, otherwise comparisons could be leading to false conclusions. This only happened in 20.6% of cases. Most studies used small temperature decrements, which allows one to move closer to the real value, although more widely spaced decrements were usually used at lower temperatures.

### 4.4. Measuring Freezing Damage

Visual assessment was the simplest method, as it requires no equipment, and thus was most commonly used (Figure 10). In VA, the samples were either visually observed and compared to a grading scale [70,109,110], or a damaged/undamaged grading was given [111,112]. Sometimes microscopes were used [113], although these instances were scored together with the rest.

Visual assessment was also the method that was most often used to observe frost damage in the field (Table 3), and, as most researchers consider field observations the gold standard, VA should be the technique that other techniques are compared to. In the studies identified in this search, VA was compared to every other technique, as was the other technique commonly used in the field, EL.

Electrolyte leakage was the second most used technique in controlled enclosure testing (Figure 10). Electrolyte leakage is a relatively simple technique, as it only requires an instrument to measure electrical conductivity, commonly available in most laboratories. Electrolyte leakage also avoids the need for grading scales that would be used according to the researcher's subjective criteria, and it is thus easier to produce results comparable between different researchers with EL. Visual assessment and EL were significantly correlated with each other, as shown in the three studies that compared the two methods [48,70,71]. Electrolyte leakage and VA were used throughout the full historical range of studies reviewed in this map (Table 7). The wide range of years across which this technique was used, its correlation with VA and the simplicity of its use, combined with well-established protocols, make this a very good technique for a researcher to use.

**Table 7.** Range of years during which each measurement technique was used.

| Technique | Years |
|---|---|
| Electrolyte leakage | 1972–2020 |
| Visual assessment | 1973–2020 |
| Fluorometry | 1990–2020 |
| Differential Thermal Analysis | 1985–2011 |
| Tetrazolium Assay | 1992–2004 |
| Electrical Impedance Spectroscopy | 1970–2017 |

Fluorometry, the third most used technique (Figure 10), was introduced in the 1990s, its use spanning between 1990 and 2020. It requires a fluorometer, and it is more complex to use. It was significantly correlated with both VA in all five studies that compared these two techniques [12,49,70,72,73]. It was also well correlated with EL [70]. Overall, fluorometry seems like a robust, well-used technique, albeit a slightly more complex one to use than EL or VA.

Differential thermal analysis, the fourth most used technique (Figure 10), ranges a wide span of years (Table 7). It is quite useful for stems, which are much harder to grade by VA, since damage is harder to estimate. The setup for DTA is quite complex, and there was no study measuring its correlation with other measurement techniques, although it seems to be more precise than VA [41]. Due to the lack of correlation studies to date, DTA should be considered a less well-proven technique.

Electrical impedance spectroscopy, which was used almost as frequently as DTA (Figure 10), is an old technique (Table 7). Electrical impedance spectroscopy and VA were correlated in one study [74], not correlated in another [75] and had similar results in another [76]. Electrical impedance spectroscopy was also correlated with EL in one study [77]. Overall, it does not seem to be an established, well-tested technique, although there was more evidence of it measuring frost damage in a similar manner to other techniques than there was for DTA.

The tetrazolium assay was a technique introduced in the 1990s, rarely used during the 12 years its use spans (Figure 10). It consists of measuring the plants' reductive potential [39,114–116]. It seems like it was an experimental technique which was briefly used for a decade and then abandoned. It was not a well-established technique, and it has not been used within the last decade.

The PM-ATPase activity measurement, which consisted of measuring plasma concentrations of the $H^+$-ATPase membrane protein [117], was only used in one study (Figure 10). It seems like another an experimental technique which has not been used in the last ten years.

## 5. Conclusions

There is a wide variety of frost damage assessment techniques, but they are all limited in what they can detect by the preceding steps. In order to obtain results that can be extrapolated to actionable information on frost tolerance in the field, the material must be grown in appropriate conditions and tested in conditions that approximate real-life events in the geographical areas where the tree is to be grown. Growth conditions need to be selected based on the objectives of the study. Freeze exposure can be achieved naturally, by waiting for natural frosts, or artificially, by inducing low temperatures with technology. Field observations can be directly used, whereas artificial freezing requires more careful extrapolation. Field observations are more costly and time consuming, as they require natural frosts. Artificial freezing is cheaper and less time consuming but needs to be carefully planned to avoid measuring something other than the desired characteristic. Frost duration, freezing rate, thawing rate, temperature range, temperature steps, and number of test temperatures should be selected to obtain the closest approximation to field results possible.

The most common techniques for measuring frost damage are VA, EL, and fluorometry. Visual assessment can be used to assess all organs but is subject to subjective interpretation. Electrolyte leakage observations are subject to size constraints, as the sample needs to be small enough to fit in a vial. Fluorometry measures the degradation of chlorophyll and can thus only be used on needles. These techniques are well correlated with each other and are widely used in the field.

**Supplementary Materials:** The following are available online at https://www.mdpi.com/article/10.3390/f12081094/s1, Table S1: Coded data.

**Author Contributions:** Conceptualization, A.-A.A.Z., K.A.S. and A.R.S.; methodology, A.-A.A.Z.; formal analysis, A.-A.A.Z.; data curation, A.-A.A.Z. and A.R.S.; writing—original draft preparation, A.-A.A.Z.; writing—review and editing, A.-A.A.Z., A.R.S. and K.A.S.; visualization, A.-A.A.Z., K.A.S. and A.R.S.; supervision, A.R.S. and K.A.S.; project administration, K.A.S. and A.R.S.; funding acquisition, K.A.S. and A.R.S. All authors have read and agreed to the published version of the manuscript.

**Funding:** Anastasia-Ainhoa Atucha Zamkova was funded by the Knowledge Economy Skills Scholarship (KESS) II, grant number 80815, and Andrew Smith was supported by the Natural Environment Research Council, grant number: NE/S015833/1. KESS is a pan-Wales higher level skills initiative led by Bangor University on behalf of the HE sector in Wales. It is part funded by the Welsh Government's European Social Fund (ESF) convergence programme for West Wales and the Valleys.

**Institutional Review Board Statement:** Not applicable.

**Informed Consent Statement:** Not applicable.

**Data Availability Statement:** Not available.

**Conflicts of Interest:** The authors declare no conflict of interest. The funders had no role in the design of the study; in the collection, analyses, or interpretation of data; in the writing of the manuscript; or in the decision to publish the results.

# Appendix A

**Table A1.** Results of exploratory search in Web of Knowledge, done on 20 November 2020.

| | Search String | Number of Hits (Web of Knowledge) | Change from Previous |
|---|---|---|---|
| 1. | **TOPIC:** ((frost OR *freezing OR subzero OR cold *) AND (toleran * OR hard * OR resistan *)) AND (gymnosperm) | 53 | |
| 2. | **TOPIC:** ((frost OR *freezing OR subzero OR cold *) AND (toleran * OR hard * OR resistan *)) AND (conifer) | 357 | Changed search word from gymnosperm to the more common name of the most common division among gymnosperms. |
| 3. | **TOPIC:** ((frost OR *freezing OR subzero OR cold *) AND (toleran * OR hard * OR resistan *)) AND (Cedrus OR Pinus OR Cathaya OR Picea OR Pseudotsuga OR Larix OR Pseudolarix OR Tsuga OR Nothotsuga OR Keteleeria OR Abies) | 1308 | Changed search word from conifer to a list of the Latin names of the most common conifer species. |
| 4. | **TOPIC:** ((frost OR *freezing OR subzero OR cold *) AND (toleran * OR hard * OR resistan *)) AND (Cedrus OR Pinus OR Cathaya OR Picea OR Pseudotsuga OR Larix OR Pseudolarix OR Tsuga OR Nothotsuga OR Keteleeria OR Abies OR cedar OR celery-pine OR cypress OR fir OR juniper OR larch OR pine OR redwood OR spruce OR yew OR softwood) | 1949 | Added common names of the most common conifer species. Added common name for conifer wood. |

**Table A1.** *Cont.*

| | Search String | Number of Hits (Web of Knowledge) | Change from Previous |
|---|---|---|---|
| 5. | **TOPIC:** ((frost OR *freezing OR subzero OR cold *) AND (toleran * OR hard * OR resistan *)) AND (Cycas OR Dioon OR Bowenia OR Macrozamia OR Lepidozamia OR Encephalartos OR Stangeria OR Ceratozamia OR Microcycas OR Zamia OR Ginkgo OR Welwitschia OR Gnetum OR Ephedra OR Cedrus OR Pinus OR Cathaya OR Picea OR Pseudotsuga OR Larix OR Pseudolarix OR Tsuga OR Nothotsuga OR Keteleeria OR Abies OR Araucaria OR Wollemia OR Agathis OR Phyllocladus OR Lepidothamnus OR Prumnopitys OR Sundacarpus OR Halocarpus OR Parasitaxus OR Lagarostrobos OR Manoao OR Saxegothaea OR Microcachrys OR Pherosphaera OR Acmopyle OR Dacrycarpus OR Dacrydium OR Falcatifolium OR Retrophyllum OR Nageia OR Afrocarpus OR Podocarpus OR Sciadopitys OR Cunninghamia OR Taiwania OR Athrotaxis OR Metasequoia OR Sequoia OR Sequoiadendron OR Cryptomeria OR Glyptostrobus OR Taxodium OR Papuacedrus OR Austrocedrus OR Libocedrus OR Pilgerodendron OR Widdringtonia OR Diselma OR Fitzroya OR Callitris OR Actinostrobus OR Neocallitropsis OR Thujopsis OR Thuja OR Fokienia OR Chamaecyparis OR Callitropsis OR Cupressus OR Juniperus OR Xanthocyparis OR Calocedrus OR Tetraclinis OR Platycladus OR Microbiota OR Austrotaxus OR Pseudotaxus OR Taxus OR Cephalotaxus OR Amentotaxus OR Torreya OR cedar OR celery-pine OR cypress OR fir OR juniper OR larch OR pine OR redwood OR spruce OR yew OR conifer OR softwood) | 2168 | Expanded list of Latin names to include all Latin names for gymnosperm species. |
| 6. | **TOPIC:** ((frost OR *freezing OR subzero OR cold *) AND (toleran * OR hard * OR resistan *)) AND (Cycas OR Dioon OR Bowenia OR Macrozamia OR Lepidozamia OR Encephalartos OR Stangeria OR Ceratozamia OR Microcycas OR Zamia OR Ginkgo OR Welwitschia OR Gnetum OR Ephedra OR Cedrus OR Pinus OR Cathaya OR Picea OR Pseudotsuga OR Larix OR Pseudolarix OR Tsuga OR Nothotsuga OR Keteleeria OR Abies OR Araucaria OR Wollemia OR Agathis OR Phyllocladus OR Lepidothamnus OR Prumnopitys OR Sundacarpus OR Halocarpus OR Parasitaxus OR Lagarostrobos OR Manoao OR Saxegothaea OR Microcachrys OR Pherosphaera OR Acmopyle OR Dacrycarpus OR Dacrydium OR Falcatifolium OR Retrophyllum OR Nageia OR Afrocarpus OR Podocarpus OR Sciadopitys OR Cunninghamia OR Taiwania OR Athrotaxis OR Metasequoia OR Sequoia OR Sequoiadendron OR Cryptomeria OR Glyptostrobus OR Taxodium OR Papuacedrus OR Austrocedrus OR Libocedrus OR Pilgerodendron OR Widdringtonia OR Diselma OR Fitzroya OR Callitris OR Actinostrobus OR Neocallitropsis OR Thujopsis OR Thuja OR Fokienia OR Chamaecyparis OR Callitropsis OR Cupressus OR Juniperus OR Xanthocyparis OR Calocedrus OR Tetraclinis OR Platycladus OR Microbiota OR Austrotaxus OR Pseudotaxus OR Taxus OR Cephalotaxus OR Amentotaxus OR TORreya OR cedar OR celery-pine OR cypress OR fir OR juniper OR larch OR pine OR redwood OR spruce OR yew OR conifer OR softwood) AND (test * OR technique * OR measure * OR treat * OR trait OR analys *) | 1483 | Added search word for techniques, as it was found that search was not specific enough. |

Table A2. Gymnosperm species included in this review and the number of studies which include them.

| Species Name | N | Species Name | N | Species Name | N |
|---|---|---|---|---|---|
| *Picea abies* | 56 | *Pinus tecumannii* | 2 | *Abies chensiensis* | 1 |
| *Pinus sylvestris* | 50 | *Podocarpus lawrenci* | 2 | *Abies grandis* | 1 |
| *Pseudotsuga menziesii* | 46 | *Abies procera* | 2 | *Abies koreana* | 1 |
| *Picea glauca* | 26 | *Abies nephlorepsis* | 2 | *Abies homolepis* | 1 |
| *Picea rubens* | 22 | *Abies holophylla* | 2 | *Tsuga dumosa* | 1 |
| *Picea mariana* | 20 | *Abies veitchii* | 2 | *Tsuga sieboldii* | 1 |
| *Pinus contorta* | 16 | *Abies nordmanniana* | 2 | *Tsuga diversifolia* | 1 |
| *Picea sitchensis* | 12 | *Abies fraseri* | 2 | *Tsuga yunnanensis* | 1 |
| *Picea engelmannii* | 9 | *Abies sachaliensis* | 2 | *Larix sukaczewii* | 1 |
| *Pinus banksiana* | 8 | *Thuja occidentalis* | 2 | *Larix sibirica* | 1 |
| *Chamaecyparis nootkatensis* | 8 | *Larix leptolepis* | 2 | *Larix gmelinii* | 1 |
| *Pinus radiata* | 7 | *Cupressocyparis leylandii* | 2 | *Larix potanini* | 1 |
| *Thuja plicata* | 7 | *Pseudotsuga sinensis* | 1 | *Larix potanini* | 1 |
| *Pinus halepensis* | 5 | *Pinus albicaulis* | 1 | *Larix occidentalis* | 1 |
| *Pinus taeda* | 5 | *Pinus densiflora* | 1 | *Larix cajanderi* | 1 |
| *Pinus strobus* | 5 | *Pinus pseudostrobus* | 1 | *Diselma archeri* | 1 |
| *Pinus resinosa* | 5 | *Pinus monticola* | 1 | *Phyllocladus aspleniifolius* | 1 |
| *Pinus nigra* | 4 | *Pinus bungeana* | 1 | *Cupressus sempervirens* | 1 |
| *Pinus ponderosa* | 4 | *Picea pungens* | 1 | *Sabina przewalskii* | 1 |
| *Pinus pinaster* | 4 | *Picea smithiana* | 1 | *Cedrus libani* | 1 |
| *Abies alba* | 4 | *Picea brachytyla* | 1 | *Cedrus deodara* | 1 |
| *Larix decidua* | 4 | *Picea likiangensis* | 1 | *Keteleeria evelyniana* | 1 |
| *Pinus cembra* | 3 | *Picea jezoensis* | 1 | *Juniperus sinensis* | 1 |
| *Pinus greggii* | 3 | *Picea glehnii* | 1 | *Juniperus recurva* | 1 |
| *Pinus oocarpa* | 3 | *Picea jezoensis* | 1 | *Agathis australis* | 1 |
| *Pinus wallichiana* | 3 | *Picea asperata* | 1 | *Agathis vicennia* | 1 |
| *Pinus elliotii* | 3 | *Podocarpus macrophyllus* | 1 | *Dacrydium colensoi* | 1 |
| *Pinus mugo* | 3 | *Podocarpus oleifolius* | 1 | *Dacrydium bidwillii* | 1 |
| *Pinus caribaea* | 3 | *Podocarpus ferrugineus* | 1 | *Dacrydium cup res sinum* | 1 |
| *Podocarpus totara* | 3 | *Podocarpus hallii* | 1 | *Dacrydium biforme* | 1 |
| *Abies lasiocarpa* | 3 | *Podocarpus salignus* | 1 | *Dacrydium laxifolium* | 1 |
| *Abies balsamea* | 3 | *Podocarpus latifolius* | 1 | *Dacrydium colensoi* | 1 |
| *Tsuga mertensiana* | 3 | *Podocarpus henkelii* | 1 | *Libocedrus plumosa* | 1 |
| *Tsuga heterophylla* | 3 | *Podocarpus nivalis* | 1 | *Libocedrus bidwillii* | 1 |
| *Larix laricina* | 3 | *Podocarpus nivalis* | 1 | *Araucaria cunninghamii* | 1 |
| *Pinus brutia* | 2 | *Abies amabilis* | 1 | *Araucaria bidwillii* | 1 |
| *Pinus canariensis* | 2 | *Abies spectabilis* | 1 | *Callitris oblonga* | 1 |
| *Pinus pinea* | 2 | *Abies ernestii* | 1 | *Athrotaxis selaginoides* | 1 |
| *Pinus hartwegii* | 2 | *Abies delavayi* | 1 | *Dacrycarpus dacrydioides* | 1 |
| *Pinus patula* | 2 | *Abies mariesii* | 1 | *Callitropsis nootkatensis* | 1 |
| *Pinus maximinoi* | 2 | *Abies firma* | 1 | | |

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
