# Peer review of "Methods for Measuring Frost Tolerance of Conifers: A Systematic Map"

_forests, doi:10.3390/f12081094_

Round 1

Reviewer 1 Report

Dear authors.

The review gives a historical overview of the methodology of freezing tests for last 40 years but nothing more. New equipment and technic is not presented and the key references are old. I don't think the review is useful for experienced researchers but might be useful as a start for new researchers. Comments are given directly in the PDF file. First two technical things, a) the no. of the reference in square brackets is often missing when using author name and year in the text and b) it should be enough to write out the name of the tree species with fully scientific name only first time the tree species is mentioned in the text and only with common name in English after that.

The introduction could give better information about practical use of freezing tests, the nature of frost damage in conifers and the difference between methods assessing the damage referring to key references. Results should be presented clear and simply in tables and figures without replication in text. The discussion is the best part of the article. The conclusion should focus more on the objectives listed in the introduction.

Author Response

Response in attached document

Reviewer 2 Report

Line 43 - the same reference cited twice in one sentence. Please remove one

Line 44 – remove one “,”

Lines 56-60 and lines 61-64 – sounds like a repetition of the same ideas. Please combine them or remove one paragraph.

Lines 131 – please clarify this statement: “283 were selected and scored” while in lines 134 “n = 264”. Were there only 264 articles used also across the text? Or 283?

Lines 193-209: it seems that there is a layout problem with Table 4. Please check it.

Line 298: should it be “K s-1” or K h-1? If K s-1 was intended, please explain in the text.

Line 305 – Remove or move to supplementary table 7, redundant information

Figure 3 – Please add in the caption “K/h - rate of temperature decrease per hour”

Line 347 – Remove or move to supplementary table 8

Figure 5 - Please add in the caption “K/h - rate of temperature decrease per hour”

Line 691 – please explain abbreviation when used the first time. “LT50 - median Lethal Time”

Please consider shortening the “Discussion” section as it tends to repeat a lot of information already provided in the “Results”.

Author Response

Response in attached file

Round 2

Reviewer 1 Report

Much better than the first version. Professional proofreading would help to make the text more concise. The review and long list of references will be helpful for new researchers in this field to plan freezing tests.